# Transcriptional Response in a Sepsis Mouse Model Reflects Transcriptional Response in Sepsis Patients

**DOI:** 10.3390/ijms23020821

**Published:** 2022-01-13

**Authors:** Florian Rosier, Nicolas Fernandez Nuñez, Magali Torres, Béatrice Loriod, Pascal Rihet, Lydie C. Pradel

**Affiliations:** TAGC, Inserm UMR1090 Aix-Marseille Université, 163 Avenue de Luminy, 13288 Marseille, France; florian.rosier91@gmail.com (F.R.); fernandez.nunez.nicolas@gmail.com (N.F.N.); magali.torres@univ-amu.fr (M.T.); beatrice.loriod@inserm.fr (B.L.)

**Keywords:** sepsis, mouse models, patients, lipopolysaccharide, microarray, monocytes/macrophages

## Abstract

Mortality due to sepsis remains unacceptably high, especially for septic shock patients. Murine models have been used to better understand pathophysiology mechanisms. However, the mouse model is still under debate. Herein we investigated the transcriptional response of mice injected with lipopolysaccharide (LPS) and compared it to either human cells stimulated in vitro with LPS or to the blood cells of septic patients. We identified a molecular signature composed of 2331 genes with an FDR median of 0%. This molecular signature is highly enriched in regulated genes in peritoneal macrophages stimulated with LPS. There is significant enrichment in several inflammatory signaling pathways, and in disease terms, such as pneumonia, sepsis, systemic inflammatory response syndrome, severe sepsis, an inflammatory disorder, immune suppression, and septic shock. A significant overlap between the genes upregulated in mouse and human cells stimulated with LPS has been demonstrated. Finally, genes upregulated in mouse cells stimulated with LPS are enriched in genes upregulated in human cells stimulated in vitro and in septic patients, who are at high risk of death. Our results support the hypothesis of common molecular and cellular mechanisms between mouse and human sepsis.

## 1. Introduction

Sepsis is defined as an organ dysfunction caused by trauma and/or infection [1]. Sepsis results from a complex dysregulation of inflammation inducing the incapacity of the host to control an infection [2,3]. An exacerbated response of the immune system injures its own tissue and induces severe sepsis (sepsis with organ dysfunction) and septic shock, the dreaded complications of this pathology. The kinetics of the inflammatory response are not well understood [4]. An initial hyperinflammatory state induces early death. Originally, an immunosuppressive response was supposed to occur after the hyperinflammatory phase of sepsis, but evidence has shown that the anti-inflammatory response occurs at the same time as the pro-inflammatory response. The problem would therefore lie in the persistent state of immunosuppression leading to an increased risk of late mortality [5,6,7,8]. Sepsis is therefore difficult to treat because it is a multifactorial disease characterized by a dynamic clinical course. The clinical outcome is variable as it depends on several factors such as the pathophysiological process, the environment, and the genetic background, as well as the co-morbidity of the affected patients [8,9]. The prevalence of the disease is higher in low- and middle-income countries [10,11]. According to the World Health Organization (WHO), sepsis affects between 47 and 50 million people per year, leading to 11 million deaths ([12], WHO Report on the burden of endemic health care-associated infection worldwide, 21 November 2017 15:11:22 2011). Sepsis can occur after any infection, including bacterial, parasitic, or viral diseases such as malaria or COVID-19. Murine models remain very useful to decipher the main causes of sepsis [13,14]. Mouse models can be classified into three groups: (i) toxin administration, (ii) administration of live pathogens, and (iii) endogenous barrier decay after cecum ligation and puncture [13]. However, the relevance of the mouse model is still under debate [13,15,16,17,18,19,20,21]. Among these different mouse models, the direct injection of toxins such as LPS into the blood, peritoneum or lung induces a strong inflammatory response that mimics the innate immune system in humans [17,21,22,23]. The advantage of this method allows technical ease and a reproducible response of the immune system with the injection of a controlled amount of toxin. LPS is a major component of most Gram-negative bacteria [23]. When mice injected with LPS are used as a model for sepsis, it is usually injected intraperitoneally into the mouse, and several studies reported similarities between this model and the human sepsis [17]. Nevertheless, the similarities of transcriptional responses in humans and mice are widely discussed [24,25]. Herein we investigated the transcriptional response in this mouse model and compared it with gene expression profiles obtained in humans [26,27,28]. Our transcriptomic study using cells extracted from the peritoneal cavity after LPS injection in mice provides evidence that the transcriptional response in mice reflects both the response of human blood cells to bacteria and the risk of mortality in patients with sepsis.

## 2. Results

### 2.1. Microarray Gene Expression Profiling

Experimental design and data analysis were carried out as shown in Figure 1. In order to identify genes modulated by LPS injection, we isolated the peritoneal cells from 8 mice during the inflammatory peak 1 h after the injection (*n* = 4) and just before death, 40 h after LPS injection (*n* = 4), and from 4 control mice that were not injected with LPS. The RNA samples were further extracted and used for microarray gene expression profiling and qPCR experiments.

As many as 62,976 probes corresponding to 39,430 coding RNAs and 16,251 LincRNAs were used to detect mRNA expression levels for each RNA sample using a SurePrint G3 Mouse Gene Expression 8 × 60 K v2 Microarray Kit, and the data were analyzed using “AgiND” library loaded into “R” for standardization. Probes with intensity values below the background noise were filtered out, resulting in 25,880 probes, which were used for differential expression analysis. Out of the 39,430 coding RNAs and 16,251 LincRNAs, the expression of 3169 probes were found to be statistically significantly different after a significance analysis of the microarrays test with an FDR median of 0%.

Thus, 3169 probes that were identified represented 2361 genes differentially expressed in sepsis, including 130 LincRNAs (Appendix A). After performing our hierarchical clustering (Figure 2), we observed that all of our samples were correctly classified. In addition, samples from LPS-injected mice (T1 and T2) were observed to be closer than those of control mice (T0). We observed a clustering of our probes, which were separated into 6 clusters after removing the duplicated probes for genes and genes that clustered in several groups. Two thousand three hundred and thirty-one genes were still differentially expressed (Appendix A). There were 109 genes in the first gene cluster, 1050 genes in cluster 2, 181 genes in the third cluster, 107 genes in the fourth cluster, 793 genes in the fifth cluster group genes, and finally 121 genes in cluster 6 (Figure 3).

We further assessed the expression of 4 selected genes (TNF, IL1b, IL6, and IL10) by RT-qPCR to confirm the expression pattern. Gene expression levels measured with a microarray and with RT-qPCR were correlated for TNF (Spearman’s rho = 0.882, *p* < 0.001), IL1b (Spearman’s rho = 0.891, *p* < 0.001), IL6 (Spearman’s rho = 0.845, *p* = 0.002), and IL10 (Spearman’s rho = 0.982, *p* < 0.001).

Figure 4 shows differences in gene expression level. It highlights that gene expression patterns using microarray and RT-qPCR methods were consistent. In particular, statistical analyses of RT-qPCR measurements confirmed that there was a higher level of gene expression at T1 for the 4 genes (*p* < 0.012).

To summarize, we identified 2361 genes that were modulated by LPS injection. RT-qPCR experiments confirmed gene expression levels for the selected genes. Gene expression profiles allowed us to correctly classify all our samples. We further analyzed the potential enrichment in functional terms to assess the biological significance of our results.

### 2.2. Functional Microarray Analysis

To analyze functional annotations related to mouse sepsis, we sought biological process gene ontology (GO) terms, mouse gene atlas terms, KEGG pathways, and disease ontology terms for the gene whose expression was up- or downregulated in mice injected with LPS (Appendix A). The analysis of functional terms showed an over-representation of terms related to signaling pathways such as TNF, IL-17, cytokine-mediated, chemokine, NF-kappa B, Toll-like receptor, NOD-like receptor, MAPK, and JAK-STAT signaling pathways (Figure 5A).

Noticeably, the functional term that yielded the best enrichment *p* value was related to the response of peritoneal macrophages to LPS (Figure 5A–C). Furthermore, several disease ontology terms were significantly over-represented. These included pneumonia, sepsis, systemic inflammatory response syndrome, severe sepsis, inflammatory disorder, immune suppression, and septic shock. Most of those functional or disease terms were found when analyzing the enrichment in individual clusters, as shown for cluster 3 and 6 in Figure 5B,C, respectively. Moreover, there was enrichment in terms that were specific for some clusters (Figure 5B,C). Also, we found that genes annotated with biological process ontology terms, such as “regulation of natural killer cell chemotaxis”, “response to interferon-gamma”, interleukin-7 mediated signaling pathway”, “regulation of cytokine production involved in inflammatory response”, and “regulation of monocyte chemotaxis”, were over-represented in cluster 6, while the TNF and NF-Kappa B signaling pathways were over-represented in cluster 3.

In conclusion, functional enrichment analyses referred in particular to physiological pathways related to a cellular response to pathogenic motifs or cytokines. Functional terms associated with inflammation or immunosuppression were also over-represented. These are expected results for a mouse model of sepsis. We further looked for such mouse genes, the orthologs of which were modulated in vitro by LPS in humans or ex vivo in septic patients.

### 2.3. Comparison of Published Lists of Human Differentially Expressed Genes with Mouse Differentially Expressed Genes

Moreover, we compared the list of 2331 differentially expressed genes in mice injected with LPS with transcriptional signatures in humans found in three papers (Appendix A). First, we compared those mouse genes with the genes regulated in primary human monocytes stimulated with LPS for 2 h or 24 h (Figure 6E,F) [27]. There was a significant overlap between the human gene list and the mouse gene list (Appendix A). We used the GSEA approach to assess the significance of the overlap between human differentially expressed genes and mouse differentially expressed genes (Appendix A). This is a robust approach, which reduces the bias due to different statistical methods and different stringency of cut-offs. Figure 6 shows that there was a significant enrichment of upregulated genes in human cells stimulated with LPS for 2 (Figure 6E) and 24 h (Figure 6F), compared to our transcriptome data, whereas there was no enrichment of downregulated human genes, compared to downregulated genes in mouse cells stimulated with LPS (Figure 7 and Appendix A). The GSEA approach allowed us to identify the leading edge gene subsets that mainly accounted for the enrichment signals: 183 and 196 genes upregulated in human cells stimulated with LPS for 2 h or 24 h were leading edge genes (Appendix A), respectively. The two leading edge gene sets were enriched in gene ontology terms related to inflammation, such as “inflammatory response”. Interestingly, *IL1B, IL6, IL10,* and *TNF* were in both gene sets.

In all, these results indicate that the response of mouse cells to LPS shares many features with that of human cells to LPS. This confirms previously reported results that the transcriptional response to sepsis is conserved in mouse and human peripheral blood mononuclear cells [28].

We also compared this published conserved gene expression profiles with the gene expression profile reported here: upregulated genes in mouse cells stimulated with LPS for 1 h were enriched in shared upregulated genes that were previously published in human sepsis due to Gram-negative bacteria infection and in mouse sepsis due to Gram-positive bacteria infection [28] (Figure 6H and Figure 7). This supports the hypothesis that transcriptional host responses to Gram-positive bacteria and Gram-negative bacteria are very similar. In the same way, Tang et al. reported that there were no signature genes that could differentiate between Gram-positive and Gram-negative sepsis in human peripheral blood mononuclear cells [29].

Moreover, upregulated genes in mouse cells stimulated with LPS for 1 h were enriched in human and mouse genes within gene sets associated with myeloid cells, whereas there was no enrichment in human and mouse genes within gene sets strongly associated with B or T lymphocytes [28] (Appendix A).

Finally, we compared the list of the differentially expressed genes in LPS injected mice with human genes whose expression was associated with mortality in patients with sepsis [26]. GSEA approach yielded a significant enrichment after multiple test correction (Appendix A, Figure 6A–D and Figure 7). Interestingly, Davenport et al. compared their lists of genes with modulated genes in human peripheral mononuclear cells stimulated with LPS, as reported [30,31]. They found 164 and 167 modulated genes in PBMC stimulated with LPS, which were up- and downregulated in susceptible patients from the discovery cohort, respectively. GSEA analysis yielded significant enrichment for upregulated genes in the discovery and validation cohorts (Figure 6A,B and Figure 7) and a significant enrichment for up- and downregulated genes in the cohort of individuals with features of endotoxin tolerance (Figure 6C,D and Figure 7). Among the 34 leading-edge upregulated genes (in the group with endotoxin tolerance) identified by the GSEA method, we identified *IL1B, IL6, IL10,* and *TNF* (Appendix A). Moreover, they were enriched in gene ontology terms related to inflammation, such as “inflammatory response”.

To summarize, genes upregulated in mouse cells stimulated with LPS were enriched both in genes upregulated in human cells stimulated with LPS and in genes upregulated in blood cells taken from susceptible septic patients. These results provide evidence of common physiological pathways between mouse and human sepsis.

### 2.4. LincRNA Study

The microarray contained 16,251 probes corresponding to lincRNAs and 4622 different lincRNAs. For each LincRNA, only the genomic position and the direction of transcription were filled in. For 12 of them, a gene symbol (BRN1-A, PINC, TUG1, HOTAIR, lincP21, NEAT2, NEAT1, NRON, H19, KCNQ1ot1, TSIX, and XIST) was specified.

We looked then for a new and better annotation of the lincRNAs. For this purpose, we identify a symbol for the differentially expressed lincRNAs using their genomic positions in Gencode. Among the 126 differentially expressed lincRNAs in mice injected with LPS (Appendix A), we identified 1 lincRNA (Malat1). Malat1 (transcription of lung adenocarcinoma associated with metastasis) had a functional annotation and has been associated with lung cancers.

To better identify the putative role of lincRNA differentially expressed in our experiments, we looked for the role of neighboring coding genes and hypothesized that nearby lincRNA regulates the expression of this gene. To investigate this, we used the GREAT software (http://bejerano.stanford.edu/great/public/html/ accessed on 25 October 2019. Our lincRNAs list was submitted to recover the genes nearby (1000 kbp on each side of our lincRNA). The significant *p*-values < 0.05 after the Benjamini correction were filtered, and the results are given in Appendix A. From 126 differentially expressed lincRNAs, approximately 208 genes were identified. Among those nearby genes, 40 coding genes were statistically differentially expressed in mice injected with LPS (Appendix A). Enrichment analysis results are shown in Appendix A and concern a list of 208 genes associated with molecular functions such as the direct DNA binding, binding to regulatory regions of DNA, and the downregulation of DNA-dependent transcription. We also noticed the enrichment of the Zinc finger proteins and motif prediction such as the GTF3A transcription factor. Of these, 21 clustered in the same group than the gene, suggesting a possible regulation of the gene regulation by the lincRNA (Appendix A). This is the case for BCL6 and Herc6 involved in the innate immune response in cluster 3 and 5, respectively, and for ASXL1 and CRABP2 involved in the regulation of retinoic acid receptor signaling pathway in clusters 2 and 5, respectively.

To conclude, we identified 126 differentially expressed lincRNAs in mice injected with LPS. Interestingly, 40 coding genes that were close to such lincRNAs were modulated by the injection of LPS. This supports the hypothesis that dysregulated lincRNAs alter the expression of nearby coding genes during the pathogenic process.

## 3. Discussion

The relevance of the mouse model has been debated for several years [3,23,32,33,34]. Seok et al. reported that transcriptional responses to acute inflammatory stresses in humans were poorly reproduced in the mouse models [35]. In contrast, Takao et al. detected a highly significant correlation between transcriptional responses in humans and those in mice using the same data sets and different analysis methods [36]. In the present study, we further investigated the genomic response to LPS in mice and compared it to that in humans.

We identified many differentially expressed genes in mouse peritoneal cells stimulated with LPS using an FDR median of 0%. Those genes were enriched in genes involved in inflammation, inflammatory disorders, immune suppression, antigen processing and presentation, apoptosis, and sepsis (pneumonia, sepsis, severe sepsis, and septic shock). The significant enrichment of differentially expressed genes were also identified in several signaling pathways, such as TNF-, IL-17, Chemokine-, NF-kappa B-, Toll-like receptor-, NOD-, MAPK-, JAK-STAT-signaling pathways. Interestingly, genes that were upregulated in cells stimulated with LPS for 1 h (Cluster 3) were enriched in genes involved in Toll-like receptor-, TNF-signaling, and NF-kappa B pathways, whereas other clusters did not show these enrichments. It should be noted that TNF and IL10 were upregulated in cells stimulated with LPS for 1 h but not in cells stimulated with LPS for 40 h. Furthermore, genes that were upregulated in cells stimulated with LPS for 1 h and 40 h (cluster 6) were enriched in genes involved in response to interferon-gamma, chemokine signaling pathway, and the regulation of chemotaxis for both natural killer cells and monocytes, whereas other clusters did not show these enrichments. The enrichment of IL-17- and JAK-STAT signaling pathways were evidenced for both clusters 3 and 6. It is likely that this reflects an early and transitional phase related to the production of TNF and a more progressive response related to the production of gamma IFN and chemokines. Noticeably, the significant enrichment of immune suppression term was found in cluster 3, indicating that this process starts quickly after stimulation with LPS. The apoptosis of immune cells has been proposed to be a key mechanism for immune suppression in sepsis patients [24,37]. Besides, IL-17 that has been shown to inhibit macrophage phagocytosis and to aggravate sepsis in a mouse model [25], likely participates in immune suppression.

In addition, our study of lincRNA located near differentially deregulated genes highlights the role of BCL6, known as an inhibitor of macrophage-mediated inflammatory responses [38,39,40] and involved in sepsis development as demonstrated in a recent publication in a mouse model [41].

It is very likely that macrophage peritoneal cells are mainly responsible for the response to LPS. Indeed, the molecular signature that yielded the best enrichment *p* value was that of the peritoneal macrophage response to LPS for the whole list of differentially expressed genes and for genes in clusters 3, 4, 5, and 6. This is also supported by the enrichment found for human monocytes stimulated by LPS for other myeloid cells but not for T and B lymphocytes. Additionally, our mouse model is restricted to a particular cell type. This is a limitation because other cell types, such as dendritic cells; T, B, and NK lymphocytes, and neutrophils are thought to be involved in the mechanisms of immune suppression [3,5].

Genes upregulated in mouse cells stimulated with LPS were enriched in genes upregulated in human PBMC and monocytes stimulated with LPS [26,27]. This confirms that the transcriptional response to LPS in mice reflects partly that of human cells. Moreover, genes upregulated in mouse cells stimulated with LPS were enriched in genes upregulated in susceptible sepsis patients [26]. Additionally, it is likely that there is a conserved sepsis molecular signature in human and mouse, as previously suggested by Godec et al. [28]. Noticeably, regulated genes in mouse cells stimulated by LPS were enriched in genes from the conserved sepsis molecular signature proposed by Godec et al. [28].

However, mouse models have their limitations due to the obvious differences between species and the type of pathogens or endotoxins injected. The use of different strains of mice with different genetic backgrounds may also alter the kinetics and intensity of the immune response, as well as the age and the gender of the mice [13,42,43]. Limitations of the LPS-induced sepsis model include the inability to explain sepsis for Gram-positive bacteria and the tendency for systemic clinical signs to appear immediately after LPS injection [15,16,17,18,19,20,21,44]. Nevertheless, we have shown that the host transcriptional responses to Gram-positive and Gram-negative bacteria are very similar, as also shown by Tang et al. [29]. Nevertheless, several teams are trying to improve animal models for studying sepsis [42,43], and there is probably a need to develop an adequate diversity of animal models that maximally recapitulate the specific phenotypes of sepsis to allow a better understanding of the pathophysiology of sepsis [13]. Besides, it is essential to extend transcriptomic studies to epigenomic, proteomic, metabolomic, microbiomic, and host–pathogen interactome studies in humans and mice to better understand sepsis etiology [45].

## 4. Conclusions

We identified 2361 genes modulated by LPS injection. The analysis of functional terms showed an over-representation of terms related to many relevant pathways and diseases, such as cytokines or chemokines signaling pathways, Toll-like receptor pathway, jack-stat pathway, antigen processing and presentation, immune suppression, sepsis, severe sepsis, or septic shock. Furthermore, our results support the hypothesis that some lincRNAs alter the expression of such genes. Besides, when comparing our list of mouse genes with the genes regulated in primary human monocytes stimulated with LPS on the one hand and with those regulated in susceptible septic patients on the other hand, we found a significant overlap. These results provide evidence of common physiological pathways between mouse and human sepsis and show that mouse models can help to better understand the pathophysiology of sepsis in humans. Moreover, mouse models can be used to establish a causal relationship between death due to sepsis on the one hand and genes and molecular pathways for which common features have been detected in humans and mice on the other. In other words, the sepsis mouse model can be used to demonstrate working hypotheses for human sepsis. More particularly, one might take advantage of our mouse model to investigate the functional role of human candidate genes and pathways in monocytes or macrophages. It should be stressed, however, that other OMICS approaches are needed in humans and mice to have a global view of etiological causes of sepsis.

## 5. Materials and Methods

### 5.1. Mice and Experimental Procedure

Twelve C57BL/6SJL mice were purchased from the Charles River Laboratory (L’Arbresle, France). C57BL/6SJL mice were housed under specific pathogen-free conditions and handled in accordance with French and European directives (EU Agreement N° A-13013 03). To model severe form of sepsis, mice were intraperitoneally injected with 200 μL (35 mg/kg) of lipopolysaccharide from *Pseudomonas aeruginosa* (L9143, Sigma-Aldrich). The mice were sacrificed by cervical rupture before injection T0 (*n* = 4) and 1 h after LPS injection during the inflammatory peak T1 (*n* = 4) or 40 h after LPS injection, just before death T2 (*n* = 4). Peritoneal cavity cells were collected by peritoneal washes using PBS.

### 5.2. RNA Isolation and cDNA Preparation for Real-Time RT-qPCR Validation

Total RNA was extracted using the RNeasy extraction mini kit (QIAGEN, Hilden, Germany) and treated with DNase I (Qiagen). The quality of RNA was confirmed using an Agilent 2100 Bioanalyzer (Agilent Technologies, Germany) with Agilent RNA 6000 Nano Chips. Samples with a RIN higher than 8 were used. The quantity was measured using Qubit RNA Assay Kits on Qubit 2.0 Fluorometer. For real-time RT-qPCR, reverse transcription reactions were performed using random hexamers and the SuperScript II reverse transcriptase (Invitrogen). Gene expression was evaluated using the SYBR green PCR master mix (PE Applied Biosystems, Foster, CA, USA) on Stratagene MX3000P (Agilent). Specific primers used are listed in Table 1.

### 5.3. Gene Expression Microarray

The labeling of RNA was done as recommended by Agilent Technologies using the One-Color Microarray-Based Gene Expression Analysis Low Input Quick Amp Labeling. The Agilent microarray slides used is the SurePrint G3 mouse G3 8 × 60 K chip harboring 62,976 probes, including 39,430 coding RNAs and 16,251 lincRNAs (long intergenic nonRNA coding). The slides were scanned with “The Agilent Microarray Scanner”. Visualization and results were retrieved using Feature Extraction software. The raw data as well as the QC report were recorded. Twelve samples were used for analysis after quality validations.

### 5.4. Statistical Analyses

The “AgiND” library loaded into R was used for standardizing microarray data. This library was developed for the diagnosis and standardization of Agilent chips. First, the raw data were transformed into log2, and then normalization was performed by the method of quantiles. A first filter was used to remove the controls, used to validate the correct hybridization. Then, a second filter was applied, allowing the removal of the probes whose signal was close to the noise. Only 25,880 probes with an expression greater than background noise were retained for all samples in a study group. For statistical analysis, we used TMeV software to perform a SAM (significance analysis of microarray) to determine our differentially expressed genes. We used the false discovery rate (FDR) approach to correct for multiple tests; we applied an FDR median of 0%. Hierarchical clustering was further performed, using Pearson correlation coefficients, for the differentially expressed genes.

For RT-qPCR data, target gene levels were normalized based on B-actin level, which did not show any difference in microarray analysis, and quantified by the 2^−ΔΔCt^ method [46]. A Kruskal–Wallis test was used to compare gene expression levels in different conditions. A Spearman’s correlation test was performed to assess the correlation between the microarray and RT-qPCR results for each gene.

### 5.5. Functional Annotation and Enrichment of Functional Terms

Functional analysis was performed with “Enrichr” to identify the biological processes and molecular function, in which genes are involved on the basis of “Gene Ontology” (GO) terms as well as the visualization of biological pathways from “KEGG” (Kyoto Encyclopedia of Genes and Genomes). We also checked the disease associated with these genes using DisGeNET, which is a discovery platform containing one of the largest publicly available collections of genes and variants associated with human diseases. The last database interrogated was the Mouse Gene Atlas from BioGPS to find cell line enrichment in mouse tissues. The *p*-value was computed using a standard statistical method used by most enrichment analysis tools: Fisher’s exact test or the hypergeometric test. This is a binomial proportion test that assumes a binomial distribution and independence for the probability of any gene belonging to any set. Then, a *p*-value using the Benjamini-Hochberg method for correction for multiple hypotheses testing was applied to select the significant terms and pathways. In addition, an enrichment analysis was performed using the KEGG database to visualize the pathways of genes, taking into account only the number of genes found. Q-values under 0.05 were considered significant. To assess the functionality of LincRNA impact by a bioinformatics approach, we used the genomic regions enrichment of annotation tool (GREAT) [47]. GREAT evaluates sets of cis-regulatory elements by assigning each element to its likely target gene(s).

### 5.6. Comparison of Published Lists of Human Differentially Expressed Genes with Mouse Differentially Expressed Genes

Our mouse gene expression analysis was compared to human datasets published by Fairfax et al. [27] and Davenport et al. [26] Gene sets published were directly used in our analysis. Table 2 summarizes the characteristics of the studies. In addition, our mouse gene expression analysis was compared to conserved patterns of gene expression in humans and mouse models triggered by sepsis [28].

Gene set enrichment analysis (GSEA) was performed using GSEA software [48]. Given the gene sets previously published, we assessed whether the members of those gene sets were randomly distributed throughout the list of mouse differentially expressed genes, which were ranked on the basis of the SAM score. Enrichment scores and the corresponding nominal *p* values were calculated. Multiple testing was controlled by calculating the FDR. An FDR under 5% was considered significant.

## Figures and Tables

**Figure 1 ijms-23-00821-f001:**
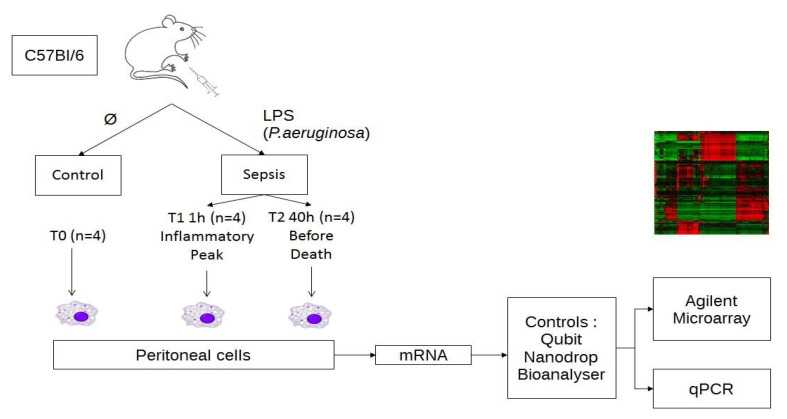
Experimental design and data analysis. Schematic outline representing the experimental design for the microarray analysis in SurePrint G3 8 × 60 K mouse from Agilent. RNA was extracted from the peritoneal cells of 4 mice (*n* = 4 for each group) injected in the peritoneal cavity either with PBS or with LPS during the inflammatory peak (T1 = 1 h) and before death (T2 = 40 h).

**Figure 2 ijms-23-00821-f002:**
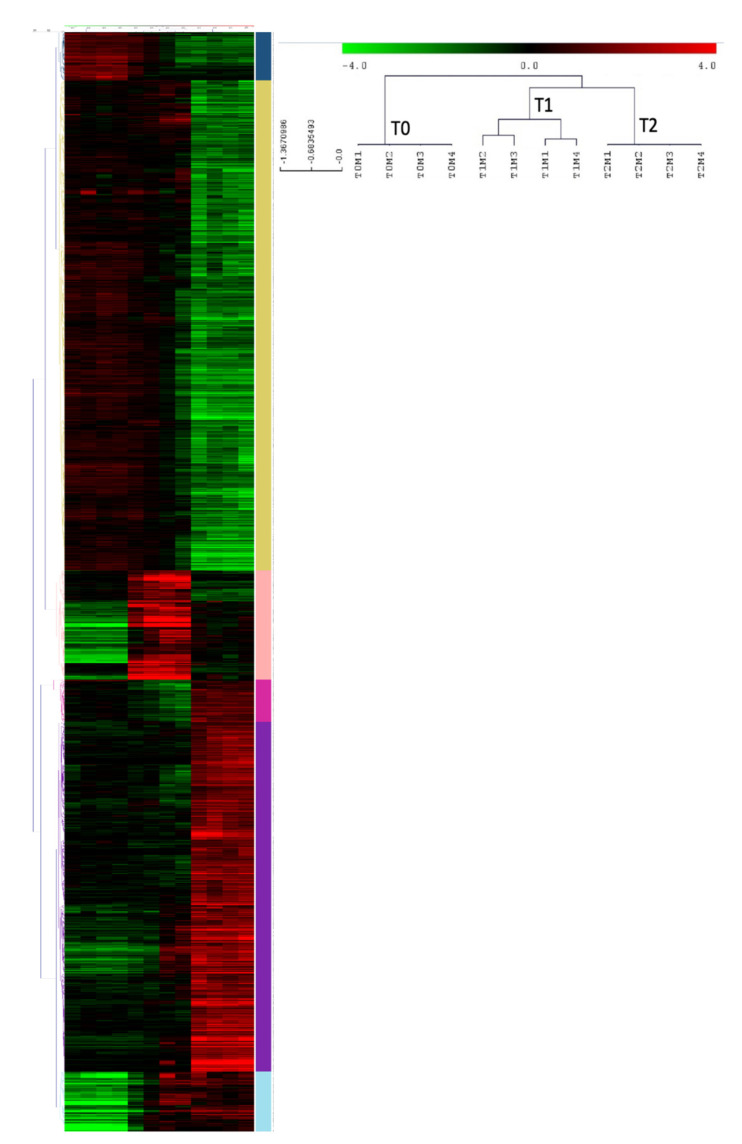
Unsupervised hierarchical clustering of probes in our mouse sepsis model. The samples in each column (vertical) and the probes in each row (horizontal) were (re)arranged. Six vertical clusters (blue, yellow, light pink, fuchsia, purple, and light blue) and 3 horizontal clusters were obtained corresponding to genes of similar expression pattern placed close to each other, and samples with comparable traits (T0, T1, and T2), respectively. The heatmap indicates upregulation (red), downregulation (green), and mean gene expression (black).

**Figure 3 ijms-23-00821-f003:**
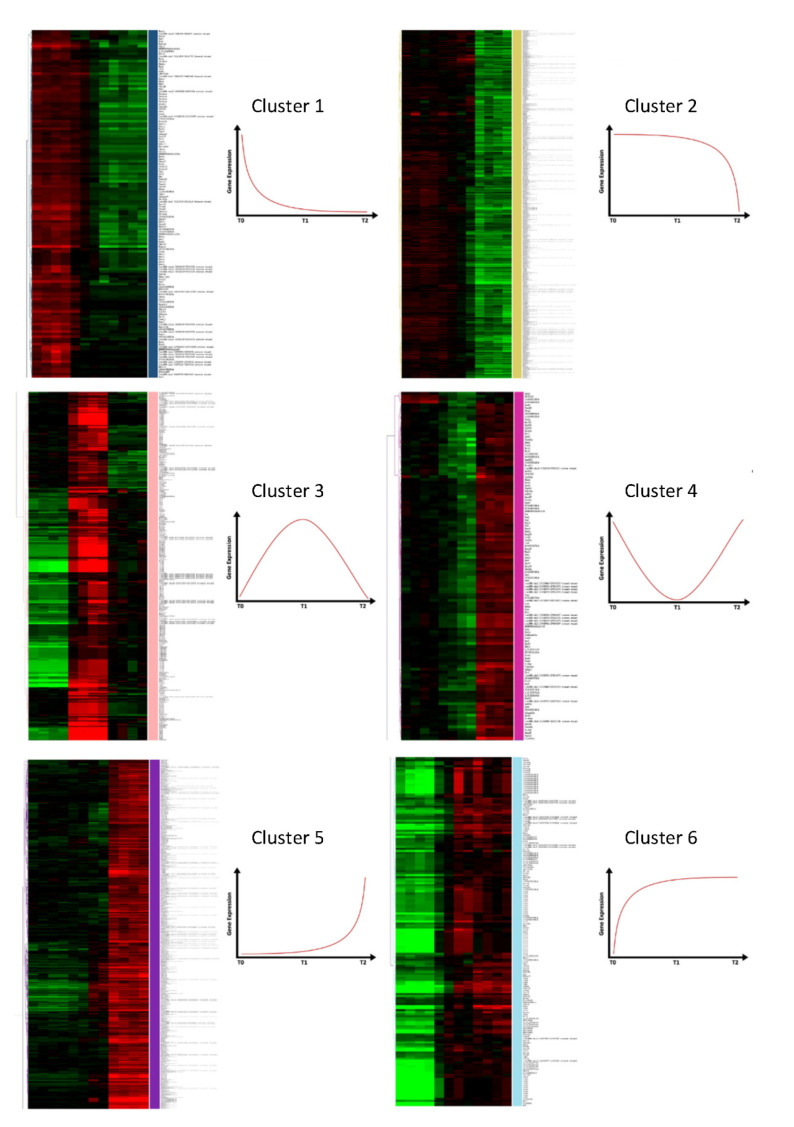
Hierarchical clustering of probes and schematic gene expression profile for each cluster identified in our mouse sepsis model (blue, yellow, light pink, fuchsia, purple, and light blue). A gene’s expression profile corresponding to each heatmap cluster is shown to its right.

**Figure 4 ijms-23-00821-f004:**
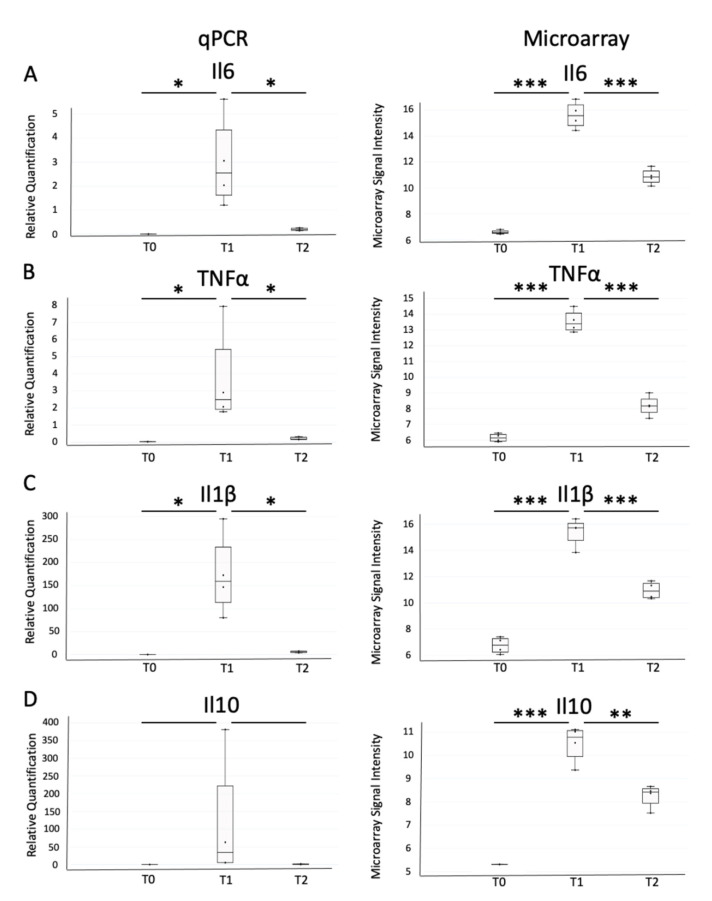
Expression levels of IL6 (**A**), TNFα (**B**), IL1b (**C**), and IL10 (**D**) measured by RT-qPCR (**left**) and with microarray (**right**). For RT-qPCR, values were normalized with the actin transcript. * *p*-value < 0.05, ** *p*-value < 0.01 and *** *p*-value < 0.001.

**Figure 5 ijms-23-00821-f005:**
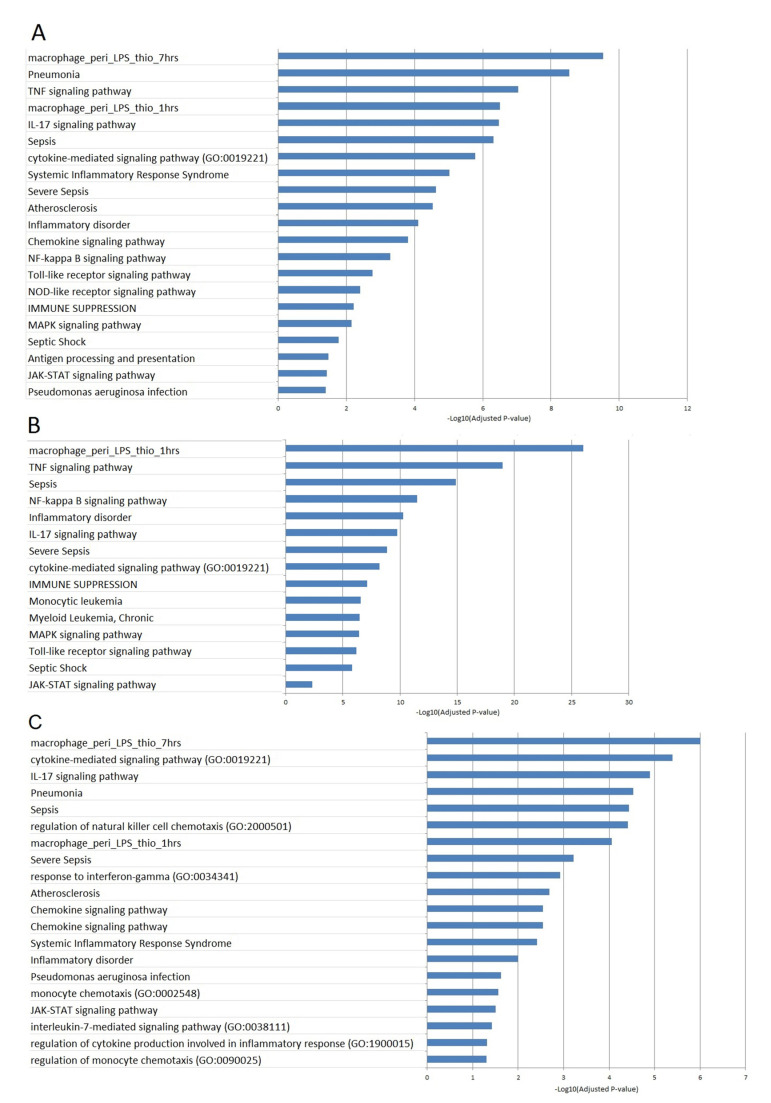
Biological processes identified by functional enrichment analysis for all clusters (**A**), cluster 3 (**B**), and cluster 6 (**C**). The negative log10 of the adjusted *p* value is represented.

**Figure 6 ijms-23-00821-f006:**
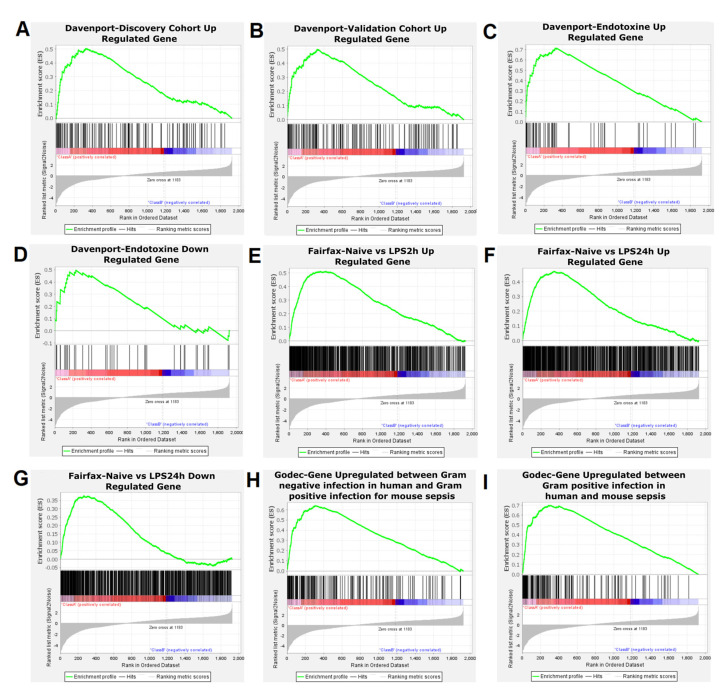
Gene set enrichment analysis (GSEA)-enrichment plots of comparison of our list of mouse differentially expressed genes with significant gene sets identified in three different papers. (**A**) Genes upregulated in the discovery cohort. (**B**) Genes upregulated in the validation cohort. (**C**) Genes upregulated in the endotoxin group (108 individuals from the discovery cohort with an immunosuppressed phenotype that included features of endotoxin tolerance). (**D**) Genes downregulated in the endotoxin group [26]. (**E**) Genes upregulated in 2 h LPS stimulated monocytes compared to naive monocytes. (**F**) Genes upregulated in 24 h LPS stimulate monocytes compared to naive monocytes. (**G**) Genes down regulated in 24 h LPS stimulated monocytes compared to 2 h LPS stimulated monocytes [27]. (**H**) Genes upregulated during Gram-negative infection in human and GRAM positive infection in mouse (GSE19668). (**I**) Genes upregulated in Gram-positive infection in human (GSE9960) or mouse (GSE19668) [28].

**Figure 7 ijms-23-00821-f007:**
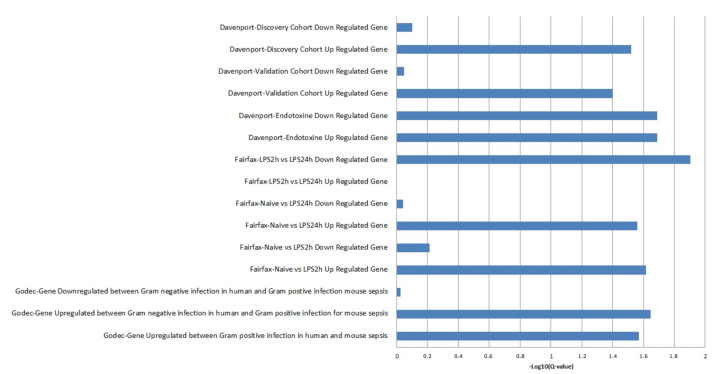
Plot of -log10 (Q-value) of gene set enrichment analysis for GSEA analysis.

**Table 1 ijms-23-00821-t001:** RT-qPCR primers sequences.

Gène	Forward Primer (5′-3′)	Reverse Primer (5′-3′)
β−Actin	TGGAATCCTGTGGCATCCATGAAACC	TAAAACGCAGCTCAGTAACAGTCCG
TNF	GGCAGGTCTACTTTGGAGTCATTGC	ACATTCGAGGCTCCAGTGAATTCGG
IL6	TGGAGTACCATAGCTACCTGGAG	TCCTTAGCCACTCCTTCTGTGACT
IL1β	GTGGTTCGAGGCCTAATAGGCT	AGCTGCTTCAGACACCTTGCA
IL10	GCCCTTTGCTATGGTGTCCTTT	TGAGCTGCTGCAGGAATGATC

**Table 2 ijms-23-00821-t002:** Characteristics of human datasets in the articles of Davenport et al. [26] and Fairfax et al. [27].

	Davenport-Discovery Cohort	Davenport-Validation Cohort	Fairfax- Naive vs. 2 h LPS Stimulated Monocytes	Fairfax- Naive vs. 24 h LPS Stimulated Monocytes	Fairfax- 2 h LPS vs. 24 h LPS Stimulated Monocytes
**Chip/Paper**	Illumina Human-HT-12 version 4 Expression BeadChips with 47,231 probes	Illumina Human-HT-12 version 4 Expression BeadChips with 47,231 probes	Illumina HumanHT-12 v4 BeadChip gene expression array platform with 47,231 probes	Illumina HumanHT-12 v4 BeadChip gene expression array platform with 47,231 probes	Illumina HumanHT-12 v4 BeadChip gene expression array platform with 47,231 probes
**Number of samples**	270 patients with sepsis due to pneumonia and organ dysfunction	114 patients with sepsis due to pneumonia and organ dysfunction	228 individuals/LPS (*E. coli*) for 2 h	228 individuals/LPS (*E. coli*) for 24 h	228 individuals/LPS (*E. coli*) for 2 h or 24 h
**Statistic**	linear regression using limma	linear regression using limma	linear regression using limma	linear regression using limma	linear regression using limma
**Cut-off**	false discovery rate of 0.05 and 1.5-fold change in expression between the two groups	false discovery rate of 0.05 and 1.5-fold change in expression between the two groups	Adjusted *p* value and cut-off fold change of +/−0.5	Adjusted *p* value and cut-off fold change of +/−0.5	Adjusted *p* value and cut-off fold change of +/−0.5
**Gene Differentially expressed**	3080 (331)	2572	2489	2085	2913
**Upregulated genes**	821 (164)	1117	1564	1116	1169
**Downregulated genes**	2260 (167)	1463	925	969	1744
**Common Up/Down**	1 (0)	8	0	0	0

## Data Availability

The transcriptomic data are available in the GEO database (GSE185150). The data used to support the findings of the study are available from the corresponding author upon request.

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
