# Peer review of "Transcriptional Response in a Sepsis Mouse Model Reflects Transcriptional Response in Sepsis Patients"

_ijms, 2022, doi:10.3390/ijms23020821_

Round 1

Reviewer 1 Report

This manuscript is very interesting and sounding,

However, I've some ocncerns about it.

Introduction is very short, and more information is needed to better focus the potential relevance of the manuscript.

A conclusion section must be added.

At last, references are really few, and only six  are from the last five years.  Please add more references, also discussing other additional model of sepsis (i.e.,  bioinformatics and exploratory data analysis tools)

Author Response

We have improved the introduction (lines 30-65), and we have added several relevant references, as requested. We have better focused the potential relevance of the manuscript. In all, we have added 22 references in the whole manuscript.

We have added a conclusion section and improved the text at the end of the manuscript (lines 348-468). In addition, we have added a conclusion in each result section. In all, this should clarify the link between our results and our conclusions.

Finally, we have written a new paragraph to discuss the relevance of mouse models, which we have mentionned in the introduction. In addition, we pointed out the interest of other OMICS studies (epigenomics, genomics, proteomics…). Integrating such high-dimensional exploratory data with transcriptomic data should led to unravel the complexity of sepsis etiology. This is the last paragraph before the conclusion (lines 333-346).

Reviewer 2 Report

Thank you for giving me the opprotunity to review the paper by Rosier et al. The paper is well written, with a detailed methology. The transferability of a mouse model in to humans can be discussed, but the results are interesting. Just the quality of nearly all figures is not sufficient. It is often not possible to identify the content.

Author Response

We have discussed the relevance of mouse models for human sepsis in a new paragraph (lines 399-412).

We have improved the resolution of somes figures.
